# Disparity and Spatial Heterogeneity of the Correlation between Street Centrality and Land Use Intensity in Jinan, China

**DOI:** 10.3390/ijerph192315558

**Published:** 2022-11-23

**Authors:** Guanwen Yin, Tianzi Liu, Yanbin Chen, Yiming Hou

**Affiliations:** College of Geography and Environment, Shandong Normal University, No. 1 Daxue Road, University Science Park, Changqing District, Jinan 250358, China

**Keywords:** street centrality, land use intensity, kernel density estimation, disparity, spatial heterogeneity

## Abstract

In the existing literature on the correlation between street centrality and land use intensity (LUI), only a few studies have explored the disparity of this correlation for different types of LUI and the differences across various locations. In response to the above shortcomings, in this study, the main urban area of Jinan, China, was taken as an example, and the disparity and spatial heterogeneity of the correlation between street centrality and LUI were explored for different categories of land use. The multiple centrality assessment (MCA) model was used to calculate the closeness centrality, betweenness centrality, and straightness centrality of the traffic network. Based on the floor area ratio (FAR) of each parcel, the utilization intensities of the residential, industrial, commercial, and public service land uses were measured. Employing the kernel density estimation (KDE) method, the street centrality of the traffic network vis-à-vis the urban LUI was rasterized into the same spatial analysis framework. The Pearson correlation coefficient and geographically weighted regression (GWR) were used to measure the correlation between the two variables and the spatial heterogeneity of the correlation, respectively. The results showed that traffic network street centrality strongly correlated with the LUI of the residential, commercial, and public service land use types, but it had a very weak association with the LUI of industrial land use. The GWR results also confirmed the spatial heterogeneity of the correlation. The results of this research highlighted the important role of traffic network street centrality in understanding the urban spatial structure. The study also helped to explain the dynamic mechanism of the road network form and the topological structure of urban spatial evolution.

## 1. Introduction

As important elements of urban systems, both traffic networks and land use affect the evolution of the urban spatial structure [1]. With the continuous growth of urbanization [2], changes in traffic networks have profoundly altered the patterns of human settlement, as well as the flow of people, goods, and vehicles, which have, in turn, changed the spatial layout and patterns of urban land use [3]. Furthermore, the reorganization of the spatial layout of cities and the corresponding changes in urban land use have led to changes in traffic demand, which have transformed the structure of traffic networks [4]. Therefore, the reciprocal relationship between traffic networks and land use has established an interactive system [5,6].

Correspondingly, analyzing the interaction and coordination between traffic networks and land use patterns have always been the focus of research in geography, planning, and urban economics. Research interests in this topic originated in the urban spatial structure models—such as concentric zones [7], sectors [8], and the multiple nuclei theory [9]—coined by the “Chicago School” [10] in the early 20th century. Nonetheless, since these seminal studies used descriptive research methods, they were greatly restricted in explaining a complex urban structure. Classical economic models and other related studies, for their part, have also emphasized the important role of the traffic network in the formation of urban spatial structures. For example, Alonso formulated the Bid Rent theory [11]. Similarly, Mills and Muth proposed the monocentric model [12,13]. Both propositions, however, rest upon the over-simplification and abstraction of urban development processes [14]. Although the economic model is indeed too limited when it comes to understanding complex real-world issues, it does emphasize the important role of transportation costs, among other parameters, in shaping urban spatial structures. In the field of urban geography and planning, the complexity of urban spatial structures has also received increasing attention [15,16,17,18,19]. Researchers have successively developed new models, such as the Garin-Lowry model [20,21] and the Wang-Guldmann model [22], to explain the interaction between traffic networks and urban land use.

In recent years, the development of network science (in particular, the spatial network) has driven the study of traffic networks, providing a theoretical tool for the in-depth analysis of the interaction between traffic systems and urban land use [23]. From this perspective, a traffic network is conceptualized as a geometric structure composed of points and lines. Over the past decades, many studies have employed this framework [24,25,26,27,28,29]. Currently, there are two popular models for studying traffic networks: space syntax and multiple centrality assessment (MCA) models. Space syntax was first proposed by Architecture Professor Bill Hillier in the 1970s [30]. It constructs a topological road network based on visibility and integration, measuring the street centrality of the traffic network through integration and choice indicators. It not only accounts for local spatial accessibility but also emphasizes overall spatial accessibility and relevance [31]. Space syntax is mostly used to explore the relationship between the street centrality of a traffic network and urban economic activities [32,33,34,35]. Building on the space syntax theory, Crucitti, Latora, & Porta proposed a multiple centrality assessment model [36,37]. This model introduces the true distance of a traffic network into the model to reflect the centrality of a node by measuring the importance of the respective node in the whole network. Compared with space syntax, the multiple centrality assessment model adopted the metric distance, which made the calculation results more reliable and realistic [38,39,40]. With the advancement of computing tools and software, the results of the multiple centrality assessment model in various research fields have been continuously enriched. In the field of urban research, the main focus is on the correlation between the street centrality of a road network and the retail locations [41,42,43,44,45] and the spatial distribution of different types of economic activities [46], employment density and the spatial distribution of the population [47], land use types [48], and land use patterns [1,49].

All this notwithstanding, there are still some areas for improvement in the scholarship on the relationship between traffic network street centrality and land use. First, existing research has mainly focused on the impact of the street centrality of a traffic network on mixed land uses [46,48]. However, the effect of traffic network street centrality on single land use categories (such as residential, commercial, or industrial land uses) is yet to be studied. Second, researchers have mainly used Points of Interest (POI) density [44], economic activity density [46], employment and population density [47] as indicators characterizing the land use intensity (LUI). Undoubtedly, such indicators can characterize the LUI to a certain extent. However, their results could be more convincing if they had used land use data. Third, researchers have mostly chosen Pearson’s correlation coefficient in their exploration of the relationship between the street centrality of a traffic network and the LUI [44,46,47,49]. The Pearson correlation coefficient reflects a linear relationship between two sets of variables, which is constant in space. However, due to the unbalanced spatial distribution of the street centrality of a traffic network and the LUI, the correlation between the two varies in different locations; that is, the correlation is spatially heterogeneous. Exploring the characteristics and influencing factors of spatial heterogeneity is crucial for formulating effective urban transportation planning and land use planning practices.

Therefore, this study takes into account the significance of the above two aspects in the process of analyzing the correlation between the street centrality of a traffic network and the LUI. First, four categories of urban land use, including residential, industrial, commercial, and public service land uses, were selected. Among the four types of land use, public service land use mainly included sites and structures with such functions as administrative, cultural, educational, sports, health, and other institutions and facilities. The floor area ratios (FAR) for all the plots within each type of land use were obtained from the urban and rural construction land monitoring database of Jinan City, which is maintained by the Jinan Housing and Urban-Rural Development Bureau. The floor area ratio is the ratio of the total usable area of a building to the total area of the plot in which the building is located. The FAR, which includes the base area, height, and the number of building floors, is an effective indicator of the LUI of built-up land. The higher the rate of floor area ratio was, the greater the utilization intensity of the built-up land was. Therefore, the FAR more truly reflects the LUI. Secondly, in this study, the geographically weighted regression (GWR) model was used to detect the spatial heterogeneity of the correlation between the traffic network street centrality and the LUI. The GWR model is a useful tool that considers the effect of local characteristics [50]. For instance, the GWR coefficient varies in correspondence with the spatial position. As such, it serves as an effective framework for detecting spatial instability. The wide application of the GWR model in social, ecological, and land-use research demonstrates this point [51,52]. The overview of the research methodology of this study is illustrated in Figure 1.

The rest of this paper is structured as follows. Part 2 introduces the study area and the source of the data used. Part 3 is an introduction to the quantitative methods and models. Part 4 and Part 5 include descriptions and discussions of the model calculation results. Part 6 summarizes the main points of this study.

## 2. Study Area and Date

### 2.1. Study Area

Located on the southeastern edge of the North China Plain, Jinan is the capital of Shandong Province (Figure 2). With a long history and rich cultural heritage, Jinan is the political, economic, cultural, technological, and educational center of Shandong Province. It is also an important transportation hub. Due to China’s reform and opening-up policy, Jinan has undergone rapid development since the 1980s. In 2021, the regional GDP of Jinan amounted to 163 billion USD, the permanent population was 9.3 million people, and the urbanization rate reached 74.2%. Affected by the topography, the main urban area of Jinan has an east-west belt shape (Figure 2), with the employment centers and residential areas clustered near the central business district (CBD). In tandem with the spatial expansion of the city, high-tech industrial zones, commercial complexes, and large-scale transportation hubs have gradually emerged around Jinan. In this context, the transportation network that connects the CBD with the surrounding agglomeration centers carries a huge volume of traffic and plays an important role in developing the Jinan metro region.

### 2.2. Street Network and Urban Land Use

High-quality urban traffic network data plays a significant role in measuring street centrality. The traffic network data for the main urban area of Jinan used in this research came from the National Platform for Common Geospatial Information Service, an integrated geographic information sharing and service portal constructed by the National Basic Geographic Information Center of China. This platform combines the geographic data on public service resources from the survey and geographic information systems departments at the national, provincial, and municipal (county) levels, as well as the relevant government departments, enterprises and institutions, social organizations, and the public. It provides authoritative, standardized, and comprehensive unified online geographic information services to various users. The traffic network data for the Jinan urban area was collected on 21 October 2021. As shown in Figure 3a, the processed traffic network consisted of 5268 nodes and 7176 edges. The road segments varied in length, ranging from 0.7 m to 2889 m, with an average length of 231.3 m. The density of the road network in the main urban area of Jinan was 2.67 km/km^2^. Roads less than 200 m in length were concentrated in the urban kernel areas within the Inner Ring Road, while roads longer than 800 m were mainly distributed on the urban fringe outside the Second Ring Road (Figure 2 and Figure 3a).

The urban land use data came from the urban and rural construction land monitoring database of Jinan City. The database is maintained by the Jinan Housing and Urban-Rural Development Bureau. Researchers and research departments with scientific needs can obtain construction land data by submitting applications to the Bureau. After accessing the database, information on all types of land use was extracted. Then the proportion of various land areas was calculated. The top four were residential land (30.47%), industrial land (17.55%), commercial land (13.02%), and public service land (12.27%). The area of these four types of land accounts for 73.31% of the construction land area in the central urban area of Jinan. Based on this, these four types of land were selected as the research objects of this paper (Figure 3b). The parcel data for each land use type included information such as the building density, floor area ratio, land use size, and building base area. 

## 3. Research Methods

### 3.1. Street Centrality Measures

Based on the Multiple Centrality Assessment (MCA) model, in this study, ArcGIS software was used as the platform, and the Urban Network Analysis Tools were used to measure the street centrality of the traffic network in the main urban area of Jinan. Per the MCA model, urban road lines were used as the edges of the network, and the road intersections or endpoints were used as nodes connecting the edges [53]. Then, the distances between the nodes along the actual network path were calculated, and, in the end, the street centrality of the traffic network was measured. Three indicators of centrality—closeness, betweenness, and straightness—were selected to measure the street centrality of the road network and reflect the accessibility, choice, and traffic efficiency of the traffic network.

(1)The closeness centrality (*C^C^*) indicated the closeness of a node to all other nodes in the traffic network, as well as the accessibility of the node in the network. It was measured using the shortest road network distance and size from a given node to all nodes. The calculation formula for the closeness centrality of node *i* was
(1)$$C_{i}^{C}=\frac{N-1}{\sum_{j=1;j\neq i}^{N}d_{ij}}\;$$
where *N* is the number of traffic network nodes, and *d_ij_* is the shortest distance between nodes *i* and *j*. In short, the closeness centrality was the reciprocal of the average distance from a node to all other nodes. The smaller the average distance was, the stronger the closeness centrality was.(2)The betweenness centrality (*C^B^*) described the importance of node *i* in the traffic network by calculating the number of times the shortest path between node pairs passed through node *i*. The betweenness centrality reflected the transfer and connection functions of the nodes in the traffic network. The greater the number of shortest paths passed through a node, the stronger the betweenness centrality was, which meant that the node played the role of a bridge or a hub transfer in the entire traffic network. The calculation of the betweenness centrality of node *i* was
(2)$$C_{i}^{B}=\frac{1}{\left(N-1\right)\left(N-2\right)}\sum_{j=1;k=1;j\neq k\neq 1}^{N}\frac{n_{jk}\left(i\right)}{n_{jk}}\;$$
where *N* is the number of traffic network nodes, *n_ik_* is the number of shortest paths between nodes *j* and *k*, and *n_jk_*(*i*) is the number of shortest paths passing through node *i* in the shortest paths between nodes *j* and *k*. Unlike the closeness centrality, the betweenness centrality could capture the special attributes of the node position; that is, it was not used as the starting point or destination of travel but rather as a necessary intermediate transition point. The betweenness centrality was an important indicator for measuring the traffic flow of the network nodes.(3)The straightness centrality (*C^S^*) was used to calculate the importance of a node by calculating the average ratio of the shortest path distance over the straight-line distance between node *i* and the other nodes. The basic principle was that when the network path distance between two nodes was close to their Euclidean distance, the communication between the two nodes was easier. The straightness centrality calculation formula of node *i* was
(3)$$C_{i}^{S}=\frac{1}{N-1}\sum_{j=1;j\neq i}^{N}\frac{d_{ij}^{Eucl}}{d_{ij}}$$
where *N* is the number of traffic network nodes, $d_{ij}^{Eucl}$ is the Euclidean distance between nodes *i* and *j*, and *d_ij_* is the shortest distance between nodes *i* and *j*. As an important indicator for measuring the efficiency of traffic networks, straightness centrality is of great significance in spatial network research.

The calculation of the centrality index was, to a large degree, affected by the search radius. Because the relationship between the street centrality of the traffic network in the entire main urban area of Jinan and the urban LUI was studied, the search radius was set to infinite, and all of the nodes in the traffic network were covered during the calculation. Finally, an index reflecting the street centrality of the traffic network in the entire main urban area was obtained.

### 3.2. Kernel Density Estimation (KDE)

To analyze the correlation between the street centrality of the traffic network and the spatial distribution of the urban LUI, it was necessary to use spatial smoothing or interpolation methods to rasterize these parameters and incorporate them into the same spatial analysis framework. In this research, the kernel density estimation (KDE) method that was integrated with the ArcGIS platform was selected to smooth the data of the street centrality of the traffic network and the urban LUI and to obtain a continuous spatial distribution map.

The KDE method was used to calculate the density of discrete points within a certain window range, which was used as the center value of the window [54,55,56,57]. The kernel density at the center of the grid was the sum of the density within the window range [58]
(4)$$\hat{f}\left(x\right)=\frac{1}{nh^{d}}\sum_{i=1}^{n}K\left(\frac{x-x_{i}}{h}\right)$$
where *K()* is the kernel density equation, *h* is the threshold, *n* is the number of points within the threshold, and *d* is the dimension of the data. For example, when *d* = 2, a common kernel density equation could be defined as
(5)$$\hat{f}\left(x\right)=\frac{1}{nh^{2}\pi }\sum_{i=1}^{n}{\left[1-\frac{{\left(x-x_{i}\right)}^{2}-{\left(y-y_{i}\right)}^{2}}{h^{2}}\right]}^{2}$$

First, in this research, the Kernel Density Tool in ArcGIS 10.8 was used to calculate the street centrality of the traffic network and the LUI for different bandwidths (default bandwidth, 1 km, 2 km, and 3 km). Then the ArcGIS band set statistics tool was used to quickly calculate the correlation between the results. The results showed that there was a high correlation between pairs. Therefore, in the correlation analysis, the choice of bandwidth had no significant impact on the calculation results. After comprehensively considering the degree of smoothness and the level of detail, in this research, the bandwidth of 1000 m was selected, and the spatial resolution of the output kernel density result map was 100 m.

### 3.3. Bivariate Moran’s I

This study uses the bivariate Moran’s *I* proposed by Anselin to explore the spatial correlation and dependence characteristics of street centrality and land use intensity. The calculation method of bivariate global Moran’s *I* [59] is
(6)$$I=\frac{\sum_{i=1}^{n}\sum_{j=1}^{n}W_{ij}\left(x_{i}-\bar{x}\right)\left(y_{j}-\bar{y}\right)}{S^{2}\sum_{i=1}^{n}\sum_{j=1}^{n}W_{ij}}$$
where *I* is the bivariate global spatial autocorrelation index, that is, the correlation between the spatial distribution of spatial variables *x* and *y* in general; *n* is the total number of space units; *W_ij_* is the spatial weight matrix established by the *K* adjacency method; *x_i_* and *y_j_* are the observed values of the independent variable and dependent variable in spatial unit *i* and *j*, respectively; *S*^2^ is the variance of all samples.

Bivariate local Moran’s *I* [59] is calculated by
(7)$$I_{i}=z_{i}\sum_{j=1}^{n}W_{ij}z_{j}$$
where *I_i_* is the local correlation between the independent variable of area *i* and the dependent variable of area *j*. z*_i_* and z*_j_* are the normalized values of the variances of the observations of spatial units *i* and *j*. Four spatial patterns can be formed based on *I_i_*: High-High cluster(H-H), that is, the independent variable of spatial unit *i* and the dependent variable of adjacent unit *j* are both large; Low-Low cluster(L-L), that is, the independent variable of spatial unit *i* and the dependent variable of adjacent unit *j* are both small; Low-Hight outlier(L-H), that is, the independent variable value of spatial unit *i* is small and the dependent variable value of adjacent unit *j* is large; High-Low outlier(H-L), that is, the independent variable value of spatial unit *i* is large while the dependent variable value of adjacent unit *j* is small. H-H and L-L indicate that the independent variable value of region *i* is positively correlated with the dependent variable of region *j*, while H-L and L-H indicate that the independent variable value of region *i* is negatively correlated with the dependent variable of region *j*, and its significance indicates whether the positive or negative spatial correlation is obvious.

### 3.4. Geographically Weighted Regression (GWR)

Unlike the global model, which maintains the homogeneity of the relationship between variables, the GWR model is a typical local model that interprets spatial data as non-stationary. As a spatial location changes, the regression coefficients of the influencing factors also change; that is, the explanatory power of influencing factors varies from location to location [50]. Therefore, the results obtained with the GWR model are more realistic. In this research, the GWR model was used to analyze the spatial heterogeneity of the influence of the street centrality of the traffic network on the LUI. The expression of the GWR model was
(8)$$y_{i}=\beta_{0}\left(u_{i},v_{i}\right)+\sum_{k}\beta_{k}\left(u_{i},v_{i}\right)x_{ik}+\varepsilon_{i}$$
where (*u_i_*,*v_i_*) represents the geographic location coordinates of the *i*-th spatial cell, *β_0_* is a constant term, *β_k_* is the regression parameter of the *i*-th spatial cell and the k-th independent variable, and *ε_i_* is the random error term of the *i*-th spatial cell, which satisfied the basic assumptions of zero mean and mutual independence for homoscedasticity.

The estimated value of the regression parameter *β* of the research cell *i* in the GWR model changed with the change of the spatial weight matrix *W*(*u_i_*,*v_i_*), which could be determined according to the distance between other cells and cell *i*. Additionally, the weighted least squares method could estimate the parameter *β*. Its expression was
(9)$$\beta \left(u_{i},v_{i}\right)={\left[X^{T}W\left(u_{i},v_{i}\right)X\right]}^{-1}X^{T}W\left(u_{i},v_{i}\right)y$$
where *β*(*u_i_*,*v_i_*) represents the estimated parameters of the model, **X** represents the matrix of the independent variable explanatory values, *y* represents the dependent variable, **X***^T^* represents the transposition operation of the matrix **X**, and **W**(*u_i_*,*v_i_*) represents the weighted spatial matrix of the model. To estimate the parameter *β*(*u_i_*,*v_i_*) in the above equation, it was necessary to choose a weight function to determine **W**(*u_i_*,*v_i_*). In this research, the Gauss function method was chosen as the weight function, which was expressed as follows:(10)$$W_{ij}=exp\left(-{\left(\frac{d_{ij}}{b}\right)}^{2}\right)$$

Here, *b* is the bandwidth, and *d_ij_* is the distance between the spatial cells *i* and *j*.

## 4. Results

### 4.1. Spatial Patterns of Street Centrality

A UNA calculation was used to obtain the node street centrality of the traffic network in the main urban area of Jinan. The centrality values of the network nodes were assigned to the roads, and the street centrality value of each road was the average of the sum of the street centrality values of the nodes at both ends of the road. The distributions of the closeness centrality, betweenness centrality, and straightness centrality of the traffic network in the main urban area of Jinan are shown in Figure 4a–c, and the KDE results of the street centrality are shown in Figure 4d–f.

As Figure 4a shows, the closeness centrality presented a clear concentric ring pattern. From the center of the city to the periphery, the street centrality of the traffic network gradually decreased. This characteristic was consistent with the law of distance decay in geography. The KDE results of the closeness centrality (Figure 4d) showed that high-value areas were mainly concentrated within the Inner Ring Road, such as the commercial shopping malls and the CBD. There were several sub-centers in the north and west of the Inner Ring Road. The average distance between the nodes in these areas and all the nodes in the traffic network was the smallest. The spatial distribution characteristics of the closeness centrality reflected that the traffic accessibility of the main urban area of Jinan was marked by a multi-center structure.

The spatial distribution of the betweenness centrality was very different from those of the closeness centrality and the straightness centrality (Figure 4b). The betweenness centrality of most roads was very low, and only some important arterial roads had high street centrality, such as Quancheng Road, Luoyuan Thoroughfare, Jiefang Road, Beiyuan Avenue, Lishan Road, and Shungeng Road. These roads served as the major thoroughfares of the main urban area, which bore most of the traffic flow inside the city. The KDE results of the betweenness centrality, as illustrated in Figure 4e, showed that the high-value area presented an east–west band shape, covering the main arterial roads of the city.

The straightness centrality presented a stronger multi-center structure than the closeness centrality did. As Figure 4c, f indicate, with the exception of the high-value distribution areas in the main urban area, the Lashan Commercial District to the west of the city center, the West Railway Station Commercial District, and the high-tech development zone in the east had higher straightness centrality. The shortest path from the network node in the above areas to any node of the traffic network and the deviation from a straight line was the smallest, and the traffic efficiency was the highest.

Figure 5 shows the frequency distributions of the three centralities of the total 7176 roads in the main urban area of Jinan. In Figure 5, the horizontal axis represents the street centrality, and the vertical axis represents the frequency of the street centrality. The frequency distribution of the three centralities had its own characteristics. It can be inferred from Figure 5 that the number of roads with low closeness centrality was relatively small, the number of roads with high closeness centrality was far greater than the number of roads with low closeness centrality, and the frequency presented an increasing power law distribution. The frequency of the betweenness centrality was more in line with the decreasing exponential distribution (Figure 5), and the frequency distribution of the straightness centrality was more aligned with the Gaussian distribution (Figure 5). This showed that the frequency of the betweenness centrality was attenuated in accordance with a specific scale, while the straightness centrality was symmetrically distributed around the average point. The frequency distribution characteristics of the three centralities were similar to those in the existing research [48].

### 4.2. Spatial Distributions of Urban LUI

The quartile method divided the residential, industrial, commercial, and public service land uses into four categories according to the floor area ratio. Figure 6a–d shows the spatial distribution of each type of land use. Figure 6e-h shows that except for the industrial land use, the intensively used parcels were mainly concentrated in the area within the Second Ring Road. Specifically, in terms of the LUI, each type of land use selected in this study had its own spatial distribution characteristics. For instance, parcels with the residential land use designation and high LUI were mainly distributed in the eastern area of Daming Lake and the CBD, showing an obvious dual-kernel structure. Parcels with the industrial land use designation and high LUI were mainly located on the urban fringe. The Luzhuang Industrial Park in the north of the city, in particular, was the main agglomeration center. There were also many smaller agglomeration centers in the northeast and southwest of the city. Parcels with the commercial land use designation and high LUI showed a multi-kernel spatial distribution pattern. In this category, the CBD was the main and largest agglomeration center, with multiple secondary agglomeration centers surrounding it. The spatial distribution of the parcels with the public service land use designation and high LUI was similar to that of the parcels with the residential land use designation, and its agglomeration scale was much larger than that of the commercial land use category.

Figure 7 shows the frequency distributions of the LUI for the residential, industrial, commercial, and public service land uses. The horizontal axis represents the LUI, and the vertical axis represents the frequency of the LUI for each category. It can be seen from Figure 7 that the frequencies of the LUI for the four land use types presented a decreasing exponential curve. This meant that the number of parcels with low LUI was large, while the number of parcels with high LUI was relatively small. The declining trends of the LUIs of the four land use categories showed a consistent pattern; that is, there was a rapid declining tail in the second half. The distribution of land use intensity of the four types conforms to the power law, which is consistent with the previous research [41,60].

### 4.3. Correlation Analysis

To analyze the correlation between the street centrality of the traffic network and the urban LUI, the ArcGIS software had to be used to process the relevant data further. First, the Feature to Point tool was used to convert the polygon layer of the residential, industrial, commercial, and public service land uses into a point layer. Then the Extract Multi Values to Points tool was used to extract the kernel density values of the street centrality and the LUI corresponding to each point. The extracted data were imported into SPSS software to calculate the Pearson correlation coefficient between the street centrality (x) and the LUI (y) for each land use category. Using the methodology employed by Wang et al. [47], in this research, the correlation coefficients were calculated for four scenarios, namely x versus y, x versus ln(y), ln(x) versus y, and ln(x) versus ln(y). The calculation results are shown in Table 1.

An analysis of Table 1 revealed that except for the correlation coefficient between the betweenness centrality and the LUI of the industrial land use, the correlation coefficients for the three scenarios passed the significance test at the 0.01 level. The street centrality of the traffic network had different effects on the LUI of each land use category. From the perspective of street centrality, the correlation coefficient between the closeness centrality and the LUI was the highest, followed by the straightness centrality. This showed that the LUI in the main urban area of Jinan had a very high correlation with the location of the land use in the traffic network and traffic efficiency. It also showed that the location was still the most effective indicator of the urban land use intensity. The correlation coefficient between the betweenness centrality and the LUI was the lowest, similar to the research results of Liu et al. [48] in the City of Wuhan. However, Porta et al.’s [41] study of Bologna, Italy, showed that the betweenness centrality and the commercial land use intensity had a high correlation. This illustrated that urban development had its own characteristics.

From the perspective of land use, the LUIs of the residential, public service, and commercial land use categories had strong correlations with the three centrality indicators. The correlation between the industrial land use intensity and the three centrality indicators was extremely weak. The main function of industrial land is to provide a place for industrial activities, which generally have the characteristics of large areas and pollutants discharged during production. At the same time, the transportation of industrial raw materials and products also needs convenient transportation. Therefore, industrial land is generally distributed in the urban fringe area with a low land price. As Figure 6 illustrates, unlike the concentrated distribution of the residential, commercial, and public service land use types with the high LUI in the urban kernel areas, the industrial land use intensity was scattered on the urban fringe where the road network was sparse, leading to an extremely weak correlation to the centrality indicators.

### 4.4. Spatial Correlation between Street Centrality and Urban LUI

The bivariate global Moran’s *I* value calculated by GeoDa software is shown in Table 2. The global Moran’s *I* is positive and has passed the significance test at the level of 1%, indicating a significant and positive spatial autocorrelation between the street centrality of the traffic network and the urban LUI. From Table 2, we can see that from the perspective of land use, the spatial correlation intensity between the LUI of residential and street centrality is the largest, followed by public service land, commercial land, and industrial land. The reason is that the areas with convenient transportation have a strong attraction for residential land. In contrast, industrial land is mostly concentrated in the peripheral areas of the city, with a sparse traffic road network and low street centrality. From the perspective of the three indicators of street centrality, the spatial correlation between closeness centrality and LUIs of different functions is the largest, followed by straightness centrality, and the spatial correlation of betweenness centrality is the smallest. High traffic accessibility is the most significant road network feature in the area with high urban LUI.

Bivariate local Moran’s *I* further explores the spatial concentration between street centrality and urban LUI. Four types of spatial distribution were shown in Figure 8. From Figure 8, it can be seen that there are similar spatial correlation characteristics between the LUIs of the residential, public service, commercial, and street centrality. Among them, the H-H type is mainly concentrated in the central area of the city, the distribution proportion of L-H and H-L types is low, and the L-L type is distributed in a larger spatial range, mainly in the peripheral area of the city. In comparison, the spatial pattern between street centrality and industrial LUI significantly differs from the other three types of land use. H-H type is mainly concentrated in industrial parks in the northwest., while the L-L type is mainly distributed in the urban fringe. This is mainly affected by industrial activities and land use characteristics.

### 4.5. Spatial Heterogeneity Explored with the GWR Model

The correlation between the street centrality of the traffic network and the urban LUI described in Section 4.3 did not take into account the impact of the spatial location. Figure 4 and Figure 6 show that the kernel density values of the street centrality and the LUI had spatial agglomeration characteristics; that is, the spatial data was non-stationary, which might lead to spatial disparity in the correlation between the centrality indicators and the LUI. For this reason, in this study, a geographically weighted regression model was used to explore the spatial heterogeneity of the correlation between the street centrality of the traffic network and the urban LUI. According to the correlation analysis conclusions described in Section 4.3, in this research, the influences of the closeness centrality, betweenness centrality, and straightness centrality on the residential, commercial, and public service land use intensities were analyzed. As Table 3 indicates, twelve GWR models in total were used. The GWR tool in the ArcGIS software was used to estimate the model parameters, and for the model bandwidth, the corrected Akaike information criterion (AICc) method was adopted. In this research, the relevant parameters of the ordinary least squares (OLS) model were also calculated as a comparison. It can be seen from Table 3 that the GWR model had a lower AICc and a higher Adjusted R^2^ than the OLS model did, indicating that the fitting result of the GWR model was significantly better than that of the OLS model. Due to space constraints, not all of the relevant parameters of the GWR model could be displayed on a map. Only the spatial distribution of the β coefficients and the local R^2^ were used to explore the spatial heterogeneity of the correlation between the street centrality of the traffic network and the urban LUI (Figure 8, Figure 9, Figure 10 and Figure 11).

Figure 9a,b shows the spatial heterogeneity of the closeness centrality affecting the residential land use intensity. It can be seen from Figure 9a that the significant positive correlation was mainly distributed in the urban kernel area. Two agglomeration centers existed east of Daming Lake and west of the CBD. There was also a strong positive correlation between the northern and southern regions of the city. The negative correlation was mainly distributed in the fringe areas of the city. The higher local R^2^ was mainly distributed to the west of the CBD and to the north of Daming Lake. Figure 9c,d illustrates the spatial heterogeneity of the betweenness centrality that affected the residential land use intensity. It can be seen from Figure 9c that the significant negative correlations were scattered in the central area of the city, which was very different from the closeness centrality. The significant positive correlation and the higher local R^2^ were concentrated in several small areas in the city center and to the south of the city. Figure 9e,f displayed the spatial heterogeneity of the straightness centrality that affected the residential land use intensity. The spatial distributions of the *β* coefficients and the local R^2^ were similar to that of the closeness centrality.

Figure 10 shows the spatial heterogeneity of the closeness centrality, betweenness centrality, and straightness centrality affecting the industrial LUI. The spatial distribution of the *β* coefficients and the local R^2^ of the closeness centrality and the straightness centrality is the same (Figure 10a,b,e,f). There are mainly 4 positive correlation high-value areas, respectively, in the southwest and east edge of the study area and near Luzhuang Industrial Park and Daxinzhuang. The negative correlation is mainly concentrated near the intersection of Xiaoqing Hebei Road and Huanggang Road. The higher local R^2^ is mainly distributed on both sides of Lanxiang Road, with Luzhuang Industrial Park as the core. Figure 10c,d shows the spatial heterogeneity of the betweenness centrality that affected the LUI of industrial. Compared with closeness and straightness, only the southwest and east edges of the study area and Luzhuang Industrial Park are the high-value areas with positive intermediary correlation, and the distribution range of negative correlation is expanded to the northwest. The high value of local R^2^ is more prominent in the west of the old commercial district.

Figure 11 shows the spatial heterogeneity of the closeness centrality, betweenness centrality, and straightness centrality affecting the commercial land use intensity. The spatial distribution patterns of the *β* coefficients and local R^2^ of the closeness centrality and straightness centrality were similar. The positive correlations were mainly concentrated in the CBD in the city center and the high-tech development zone in the east of the city. The higher local R^2^ was mainly distributed in the CBD, the eastern part of the city, and the southwest area. Figure 11c,d shows the spatial heterogeneity of the betweenness centrality that affected the commercial land use intensity. The negative correlations were mainly distributed in the CBD of the city, and there was a small-scale distribution in the eastern part of the city. With no significant pattern, the distribution of the higher local R^2^ had only a small agglomeration in the southwest part of the city.

Figure 12 shows the spatial heterogeneity of the three centrality indicators affecting the public service land use intensity. Similar to the commercial land use, the spatial distribution of the *β* coefficients and the local R^2^ of the closeness centrality and the straightness centrality for this land use category tended to be the same (Figure 12a,b,e,f). The cases with negative correlations were mainly distributed in the CBD and on the urban fringe, while those with positive correlations and the higher local R^2^ were mainly concentrated in the eastern part of Daming Lake, the western part of the CBD, and the southern part of the city. Figure 12c,d shows the spatial heterogeneity of the betweenness centrality affecting the public service land use intensity. Parcels with significant positive correlations were mainly distributed in the southern and southeastern regions of the city. The higher local R^2^ did not show a clear spatial pattern for this type of land use, and the distribution pattern tended to be random.

## 5. Discussion

The findings described in Section 4.3 and Section 4.4 showed that, regardless of the land use category, there was a strong correlation between the street centrality of the traffic network and the urban LUI. This implies that centrality was a very important factor (although not the only factor) that affected the urban LUI. This conclusion is consistent with previous studies [41,46,47].

There were significant disparities in the Pearson correlation coefficients between the three centrality indicators, i.e., the closeness centrality, betweenness centrality, and straightness centrality, and the LUI of the four land use types, namely the residential, industrial, commercial, and public service land uses. The correlation coefficients between the three centrality indicators and the residential land use intensity were significantly higher than those of other land use types. The correlation coefficient between the closeness centrality and the residential land use intensity was the highest, as shown in Table 1. Two factors could explain these patterns. First, to meet the needs of urban residents concerning commuting, transportation, and supply of goods and services, investors and real estate developers compete to obtain land with prime location and convenient accessibility for residential development. Second, with the continuous increase in the population size of Jinan and the gradual growth of land prices, the intensity of residential properties increased, as evidenced by the development of high-rise residential complexes in the City. These two factors led to the development of residential properties with high traffic accessibility and high floor area ratios. In contrast, various commercial facilities in European cities have a strong correlation with the street centrality of the road networks, and the street centrality of the commercial land use is generally stronger than that of other land use types, such as residential land use [41,46]. The traffic orientation of residential land use in the main urban area of Jinan is unique for China’s big cities. The correlation coefficient between the commercial land use intensity and the closeness centrality was higher than the correlation coefficients between the commercial land use intensity and both the straight centrality and the betweenness centrality. The commercial land use intensity depended on the location of the land. Properties with convenient transportation connectivity and high accessibility links undoubtedly attracted commercial agglomerations and yielded high-intensity development patterns. The civic functions of Public service land required that these properties be located in areas with good accessibility and high traffic efficiency. Therefore, as shown in Table 1, the public service land uses had the strongest correlation with closeness centrality and straightness centrality. In addition, in urban planning, the land demand of the public service land was met first due to its public welfare nature, and areas with high accessibility were first given to such land. The correlation between industrial land use and the three centrality indicators was relatively weak. A very important reason for this was the influence of industrial suburbanization. Due to environmental and land use policies and rising land prices, a large number of industrial enterprises moved out of the city’s central areas since the 1990s, and new agglomeration centers began to form in the suburbs with sparse traffic networks. The sparse traffic network and the isolated and scattered industrial layout led to a low correlation between street centrality and industrial land use intensity.

In this study, the GWR model was used to identify and analyze the local characteristics of the centrality and the LUI. The results showed that, at the local level, the closeness centrality, betweenness centrality, and straightness centrality had significant spatial heterogeneity in terms of the impact of the LUI of four urban land use types. Specifically, the street centrality and the LUI are spatially non-stationary. While the relationship between the two parameters varied in different parts of the study area, it was consistent locally, similar to the research results of Liu et al. [48] in the City of Wuhan. The spatial heterogeneity of the correlation came from the particularity of the traffic network and the varying urban land use intensity in different locations, and was essentially the spatial differentiation of the urban system in the evolution process.

To summarize, the layout and the development intensity of the urban land for different functions had strong traffic directivity. Therefore, it was particularly necessary to analyze the traffic utilization efficiency and to evaluate the existing and planned road network from the perspective of urban functional accessibility. This work could provide some suggestions for urban land use and urban transportation planning. For example, the high-intensity utilization of residential and public service land uses in the main urban area of Jinan drove up the travel demand and traffic flow in the area. Therefore, the main urban traffic network could be adjusted from the road network topology and street centrality to reduce the vehicle capacity of key road sections and to relieve traffic pressure.

## 6. Conclusions

Based on the multiple centrality assessment model and kernel density estimation methods, in this study, the statistical characteristics and spatial distribution characteristics of the closeness centrality, betweenness centrality, and straightness centrality of the traffic network in the main urban area of Jinan were scrutinized. The Pearson correlation coefficient and the GWR model were also used to explore the relationship between the street centrality of the traffic network and the LUI of four land use categories, as well as the spatial heterogeneity of the relationship. The results confirmed the conclusions of the existing research in that the street centrality of the traffic network was strongly correlated with the urban LUI. At the same time, the results also revealed the spatial disparity of the correlation between the three centrality indicators and the intensities of different land use types.

Based on the statistical characteristics, the closeness centrality, betweenness centrality, and straightness centrality indicators of the 7176 roads in the main urban area of Jinan showed a power-law distribution, negative exponential distribution, and Gaussian distribution, respectively. From the perspective of the spatial distribution characteristics, the closeness centrality and the straightness centrality showed multi-center characteristics, which meant that travel convenience and traffic efficiency met the travel needs of residents in different locations. The single center feature of the betweenness centrality indicated that the traffic flow was concentrated in the urban kernel area within the Inner Ring Road, which increased the traffic pressure in the city center. The different land uses in the central area of Jinan city showed obvious traffic directivity. In areas with high street centrality, the intensities of the residential, commercial, and public service land uses were also very high. The closeness centrality and straightness centrality had a greater impact on the LUI than the betweenness centrality. Because it was affected by the rent bidding mechanism and residential housing choices, the intensity of the residential land use had a stronger correlation with the street centrality of the traffic network than the intensities of the commercial and public service land uses had with the street centrality of the traffic network. The industrial suburbanization process caused industrial land use to be mainly distributed in urban fringe areas where the road network was sparse, far away from the main road network of the city, and, as a result, the correlation between the LUI of this land use type and the street centrality of the road network was weak. The maps of the *β* coefficients and the local R^2^ obtained from the geographically weighted regression model showed a spatial heterogeneity in the correlation between the street centrality of the road network and the intensity of the land use. For example, the positive correlations between the closeness centrality and the residential land use intensity were concentrated in the urban kernel area. In contrast, a large number of positive correlations between the betweenness centrality and the commercial land use intensity were distributed on the urban fringe. The conclusion of this study could help explain the dynamic mechanism of the road network form and the topological structure of the urban spatial evolution, and it could provide effective support for the simulation and prediction of urban land use change.

In this study, a more detailed exploration of the correlation between the urban LUI and the street centrality of the traffic network was performed. This exploration needs improvement in terms of the following aspects. When calculating the centrality indicators using the multiple centrality assessment model, factors such as road grade and width were not considered. In fact, different grades of roads had different roles in the road network, which may have affected the results to some extent. In addition, only four types of land use, including residential, industrial, commercial, and public service land uses, were selected and analyzed in this study. Follow-up studies may consider exploring the correlation between the street centrality of the road network and the intensities of other land uses, such as logistics and storage land use, green fields, and public facilities land use, to explore the complex relationship between the traffic network and urban land use patterns.

## Figures and Tables

**Figure 1 ijerph-19-15558-f001:**
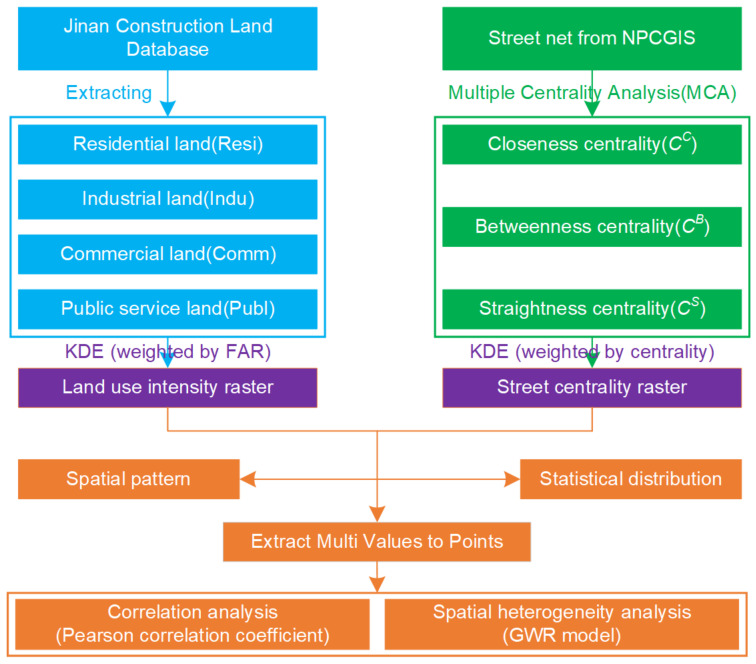
Flow chart of the current study.

**Figure 2 ijerph-19-15558-f002:**
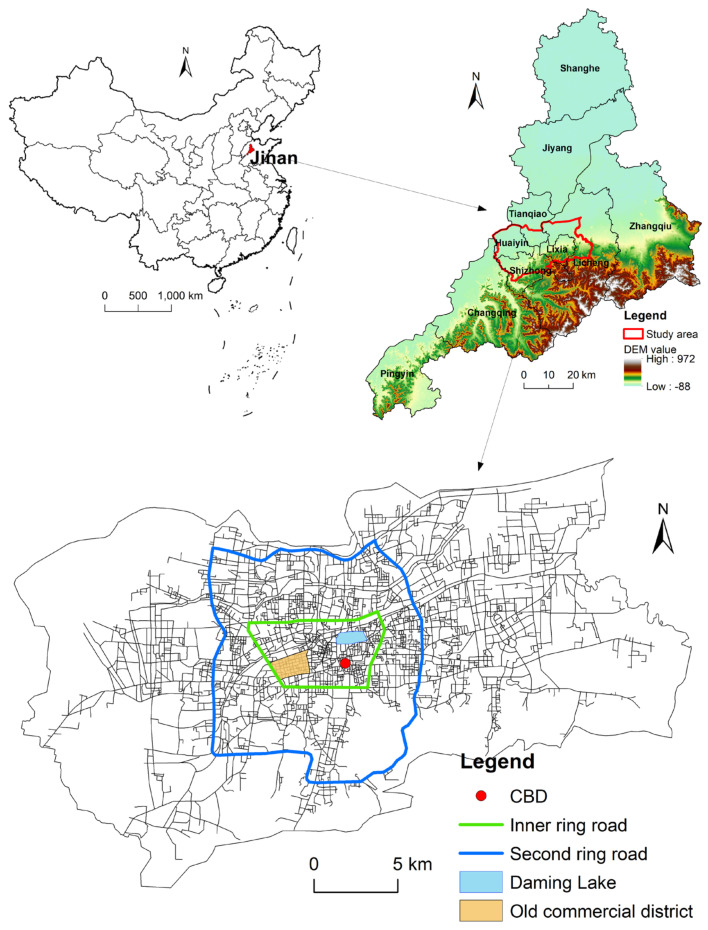
Study area.

**Figure 3 ijerph-19-15558-f003:**
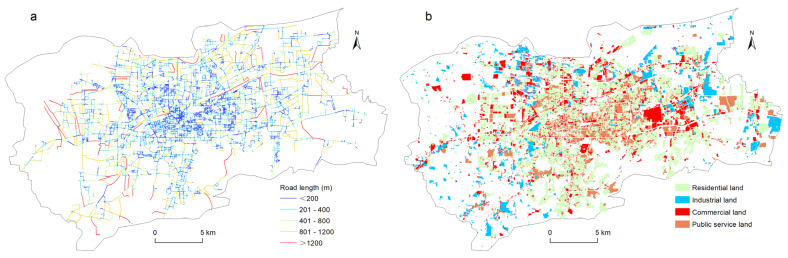
Distribution of street network (**a**) and land use (**b**) in the study area.

**Figure 4 ijerph-19-15558-f004:**
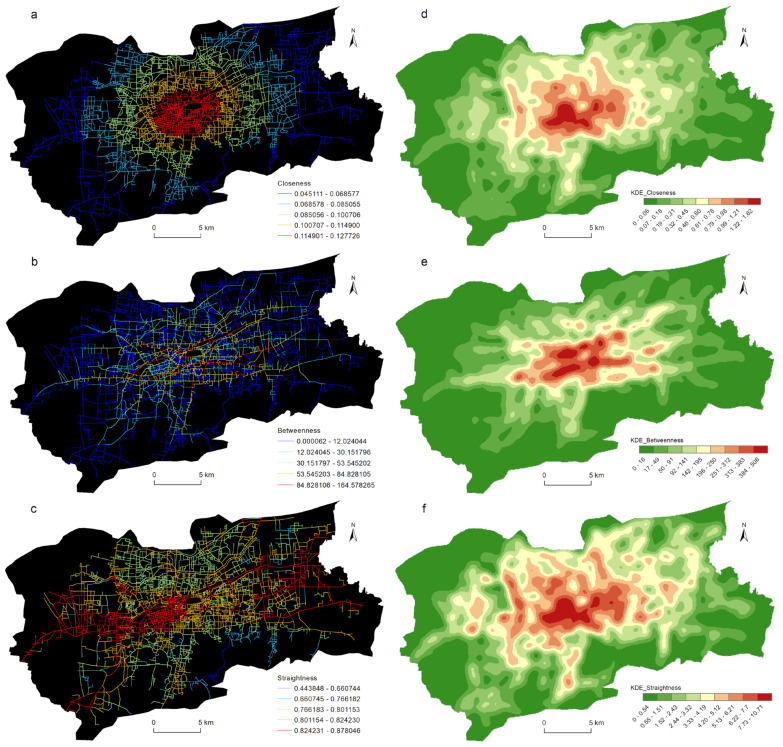
Spatial distributions of centrality indicators (**a**–**c**) and KDE of the centrality indicators (**d**–**f**).

**Figure 5 ijerph-19-15558-f005:**
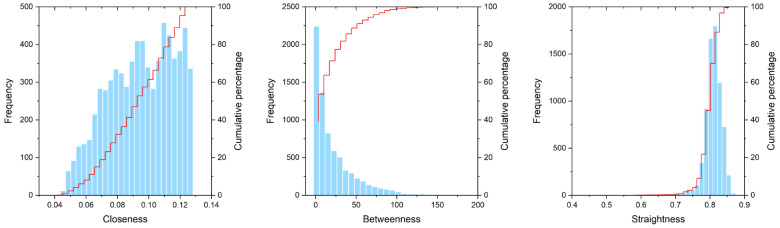
Frequency distribution of centrality indicators. The red line is the cumulative percentage curve.

**Figure 6 ijerph-19-15558-f006:**
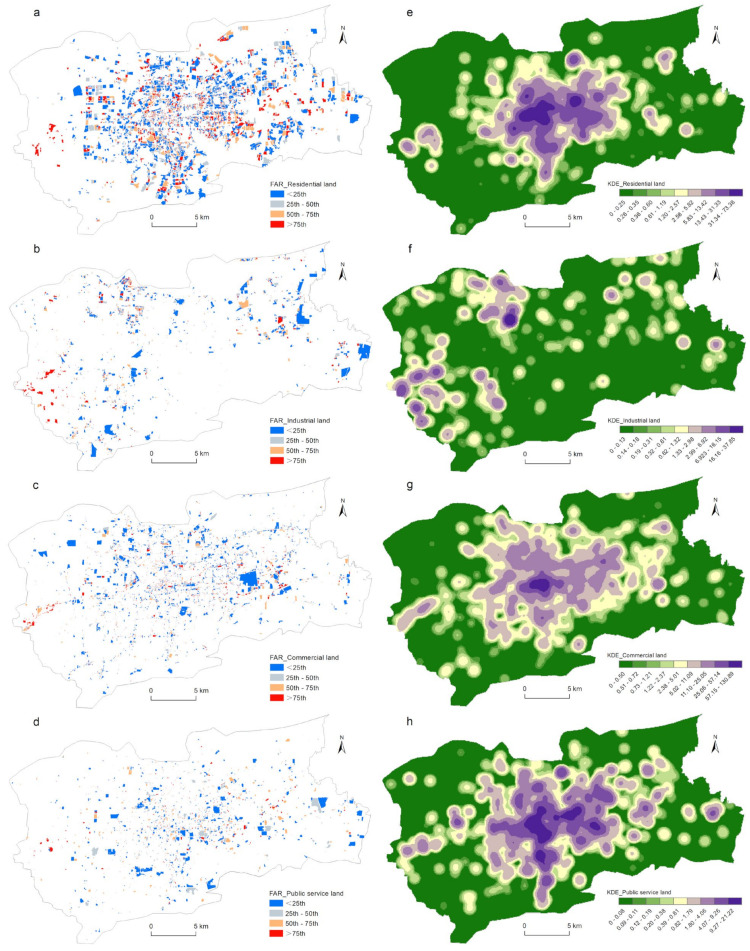
Spatial distributions of land use intensities (**a**–**d**) and KDE of land use intensities (**e**–**h**).

**Figure 7 ijerph-19-15558-f007:**
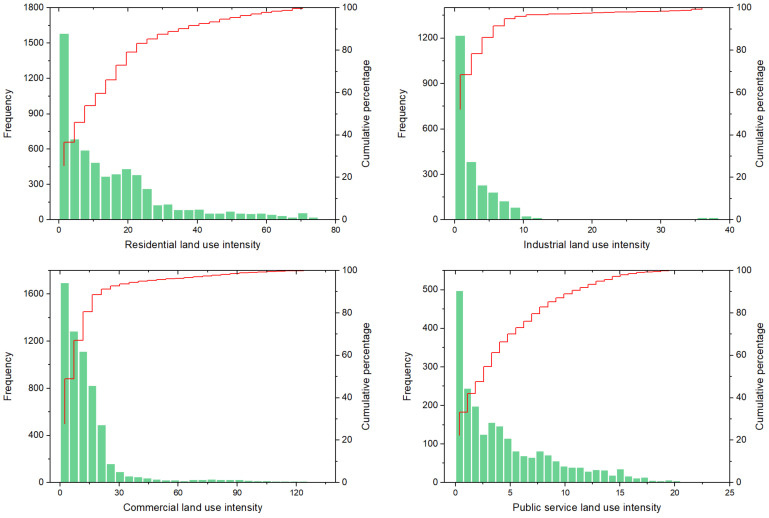
Frequency distributions of land use intensities. The red line is the cumulative percentage curve.

**Figure 8 ijerph-19-15558-f008:**
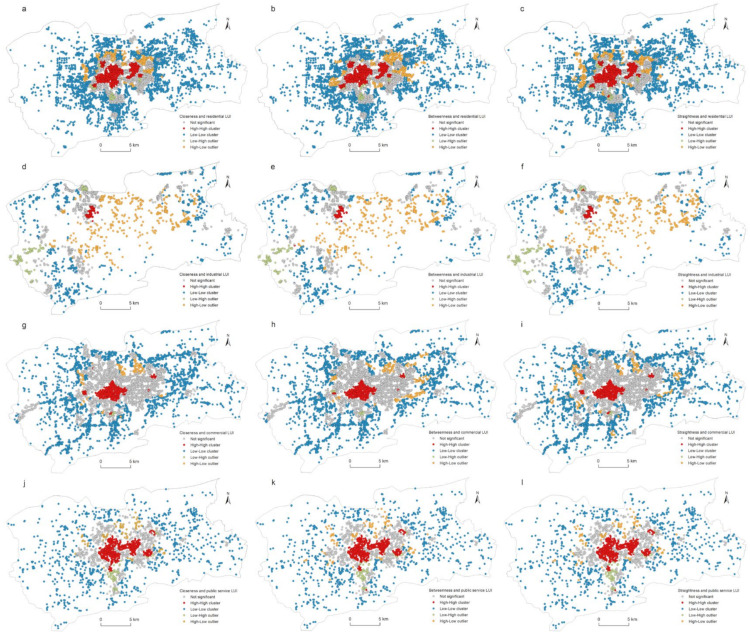
Four spatial patterns of street centrality and Urban LUI: (**a**–**c**) Closeness, betweenness straightness and residencial LUI; (**b**–**f**) Closeness, betweenness straightness and industrial LUI; (g–i) Closeness, betweenness straightness and commercial LUI; (**j**–**l**) Closeness, betweenness straightness and public service LUI.

**Figure 9 ijerph-19-15558-f009:**
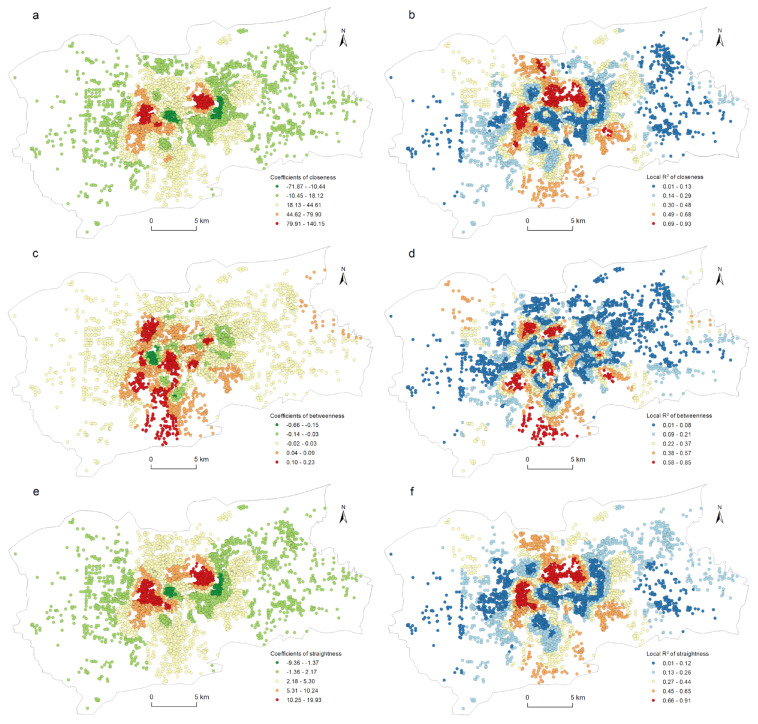
Spatial heterogeneity of the regression outputs from the GWR model for centrality and residential land use density: (a) β coefficients for closeness; (b) local R^2^ for closeness; (c) β coefficients for betweenness; (d) local R^2^ for betweenness; (e) β coefficients for straightness; (f) local R^2^ for straightness.

**Figure 10 ijerph-19-15558-f010:**
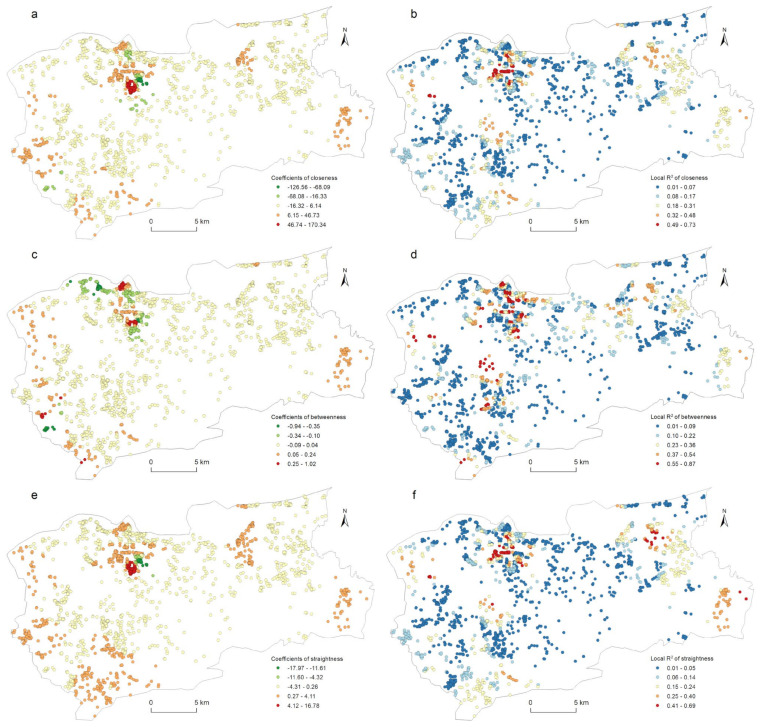
Spatial heterogeneity of the regression outputs from the GWR model for centrality and industrial land use density: (a) β coefficients for closeness; (b) local R2 for closeness; (c) β coefficients for betweenness; (d) local R2 for betweenness; (e) β coefficients for straightness; (f) local R2 for straightness.

**Figure 11 ijerph-19-15558-f011:**
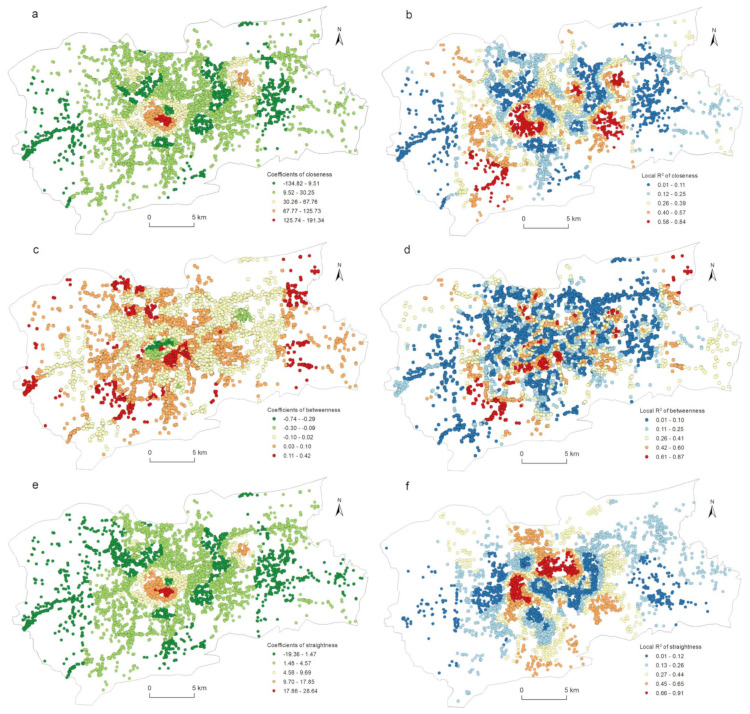
Spatial heterogeneity of the regression outputs from the GWR model for centrality and commercial land use density: (a) β coefficients for closeness; (b) local R^2^ for closeness; (c) β coefficients for betweenness; (d) local R^2^ for betweenness; (e) β coefficients for straightness; (f) local R^2^ for straightness.

**Figure 12 ijerph-19-15558-f012:**
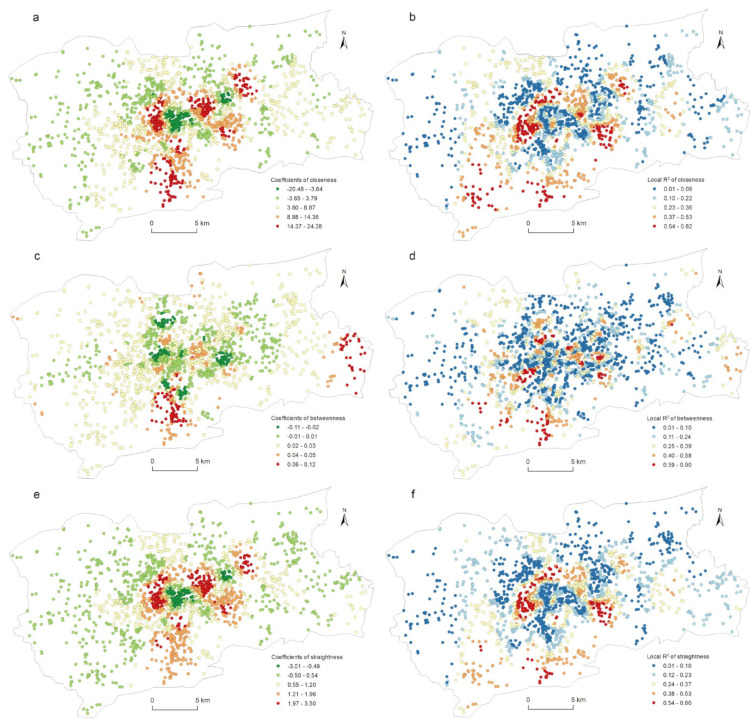
Spatial heterogeneity of the regression outputs from the GWR model for centrality and public service land use density: (a) β coefficients for closeness; (b) local R^2^ for closeness; (c) β coefficients for betweenness; (d) local R^2^ for betweenness; (e) β coefficients for straightness; (f) local R^2^ for straightness.

**Table 1 ijerph-19-15558-t001:** Pearson’s correlations between street centrality and land use intensities.

Centralities	Resi	Indu	Comm	Publ	ln(Resi)	ln(Indu)	ln(Comm)	ln(Publ)
** *C^C^* **	0.7293 ***	0.1751 ***	0.6835 ***	0.7164 ***	0.7440 ***	0.0046	0.7158 ***	0.6966 ***
** *C^B^* **	0.6537 ***	−0.0170	0.5474 ***	0.6414 ***	0.6693 ***	−0.2397 ***	0.6060 ***	0.6252 ***
** *C^S^* **	0.7077 ***	0.1733 ***	0.6662 ***	0.6911 ***	0.7267 ***	0.0676 ***	0.7157 ***	0.6938 ***
**ln(*C^C^*)**	0.5057 ***	0.1363 ***	0.4386 ***	0.5522 ***	0.6982 ***	0.0809 ***	0.6424 ***	0.6855 ***
**ln(*C^B^*)**	0.4819 ***	0.1116 ***	0.3928 ***	0.5232 ***	0.6951 ***	0.0034	0.6225 ***	0.6869 ***
**ln(*C^S^*)**	0.4584 ***	0.1170 ***	0.3893 ***	0.5003 ***	0.6302 ***	0.0873 ***	0.5799 ***	0.6306 ***

Note: *** *p* < 0.01.

**Table 2 ijerph-19-15558-t002:** Bivariate Global Moran’s *I* of street centrality and Urban LUI.

Variables	Bivariate Global Moran’s *I*	*z*-Value	*p*-Value
Closeness and Residential LUI	0.7263	217.6262	0.001
Betweenness and Residential LUI	0.6581	198.2794	0.001
Straightness and Residential LUI	0.7020	213.8111	0.001
Closeness and Industrial LUI	0.1815	32.2781	0.001
Betweenness and Industrial LUI	0.0357	6.4741	0.001
Straightness and Industrial LUI	0.1743	30.9267	0.001
Closeness and Commercial LUI	0.6797	199.9732	0.001
Betweenness and Commercial LUI	0.5516	171.3770	0.001
Straightness and Commercial LUI	0.6577	195.0182	0.001
Closeness and Public service LUI	0.7110	104.1247	0.001
Betweenness and Public service LUI	0.6527	98.2945	0.001
Straightness and Public service LUI	0.6776	99.7421	0.001

**Table 3 ijerph-19-15558-t003:** Descriptive statistics of the GWR and OLS model.

Dependent Variable	Explanatory Variables	OLS		GWR		GWR-OLS	
		AICc	Adjusted *R*^2^	AICc	Adjusted *R*^2^	AICc	Adjusted *R*^2^
Residential LUI	*C^C^*	47,542	0.53	37,790	0.90	−9752	0.37
Residential LSU	*C^B^*	48,787	0.43	35,181	0.93	−13,606	0.50
Residential LSU	*C^S^*	47,956	0.50	39,006	0.88	−8950	0.38
Industrial LUI	*C^C^*	14,625	0.03	10,989	0.80	−3636	0.77
Industrial LUI	*C^B^*	14,701	0.00	8584	0.93	−6117	0.93
Industrial LUI	*C^S^*	14,629	0.03	11,394	0.76	−3252	0.73
Commercial LUI	*C^C^*	48,901	0.47	39,642	0.88	−9259	0.41
Commercial LUI	*C^B^*	50,568	0.30	33,483	0.96	−17,085	0.66
Commercial LUI	*C^S^*	49,200	0.44	39,917	0.88	−9283	0.44
Public service LUI	*C^C^*	11,424	0.52	8763	0.86	−2661	0.34
Public service LUI	*C^B^*	11,843	0.42	7102	0.93	−4741	0.51
Public service LUI	*C^S^*	11,596	0.48	8843	0.85	−2753	0.37

## Data Availability

Not applicable.

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
