# Peer review of "Disparity and Spatial Heterogeneity of the Correlation between Street Centrality and Land Use Intensity in Jinan, China"

_ijerph, 2022, doi:10.3390/ijerph192315558_

Round 1

Reviewer 1 Report

The authors solve a very interesting problem with a novel approach.

An analysis of the spatial correlation instead of Pearson's correlation is missing. This analysis should be added to complete the study.

The analysis is correct and the results are well presented.

There is a lack of literature concerning the KDE.

Do the authors know if similar works exist to compare the results? This part should be revised and completed.

Author Response

Response to Reviewer 1 Comments

Point 1: An analysis of the spatial correlation instead of Pearson's correlation is missing. This analysis should be added to complete the study.

Response 1: Thank you for your valuable comments. We supplemented the content of spatial correlation between street centrality and urban land use intensity, using Bivariate Moran's I index. (Lines 269-294, 446-475 in manuscript)

For details, please refer to the Word document.

Point 2: There is a lack of literature concerning the KDE.

Response 2: Thanks very much for your valuable comments. We searched the relevant literature on KDE and cited it in the manuscript. The cited literature has also been added to the references of the manuscript. (Lines 252-254, lines 716-717, line 759, line 780 in manuscript)

Point 3: Do the authors know if similar works exist to compare the results? This part should be revised and completed.

Response 3: Thank you very much for the suggestion. We carefully read the existing similar literature and compared it with our research conclusions. The comparison with existing research results has been supplemented. (Lines 372-373, lines 401-403, lines 428-433, lines 569-570, lines 585-588, line 614 in manuscript)

For details, please refer to the Word document.

Reviewer 2 Report

This paper discusses Disparity and Spatial Heterogeneity of the Correlation Between Street Centrality and Land Use Intensity, and it can help to explain the dynamic mechanism of the road network form and the topological structure of urban spatial evolution. Before it is in good shape to be published, I think it should be consolidated and improved further in the future.

Firstly, is it effective to reflect land use density with floor area ratio? Whether the floor area ratio is applicable to each type of land use intensity. Whether the population density and building density can be combined to reflect the land use density.

Secondly, when considering the land use intensity of industrial land and the centrality of traffic network, whether the particularity of industrial land should be considered.

Third, in the study of spatial heterogeneity, according to the correlation study, whether it is unreasonable to consider the reason why industrial land use does not participate in the analysis of spatial heterogeneity, because there may be inconsistency between space and property.

In addition, the paper does not explain why it is necessary or meaningful to choose four types of land use, and the dynamic mechanism between traffic network and land use intensity can be supplemented.

Finally, in line 390,four scenarios should be replaced by three scenarios, because the line 388 mention except the industrial land use.

Author Response

Response to Reviewer 2 Comments

Point 1: Firstly, is it effective to reflect land use density with floor area ratio? Whether the floor area ratio is applicable to each type of land use intensity. Whether the population density and building density can be combined to reflect the land use density.

Response 1: Thank you for your valuable comments. Both floor area ratio and building density are effective indicators to reflect various land use intensity. Floor area ratio refers to the ratio of the total building area above the ground to the area of a parcel of land. Under reasonable space environment conditions, the larger the floor area ratio is, the higher the land use intensity is. The relationship between floor area ratio(FAR), building density(BD) and building floors(BF) can be expressed as FAR=BD×BF. So the floor area ratio comprehensively considers the influence of building density and building height, reflecting the use of land in three-dimensional space. Based on the above considerations, we selected the floor area ratio as the indicator to reflect the land use intensity.

It is a valuable idea to combine population density and building density as indicator of land use intensity, which takes into account both urban land development intensity and population activity intensity. This provides a very valuable idea for our follow-up research. Thanks again for your valuable suggestion.

Point 2: Secondly, when considering the land use intensity of industrial land and the centrality of traffic network, whether the particularity of industrial land should be considered.

Response 2: Thanks a lot for your valuable suggestion. According to your suggestion, in Section 4.3, when analyzing the relationship between the centrality of traffic network and the intensity of industrial land, we added the content about the particularity of industrial land. (Lines 437-441)

Point 3: Third, in the study of spatial heterogeneity, according to the correlation study, whether it is unreasonable to consider the reason why industrial land use does not participate in the analysis of spatial heterogeneity, because there may be inconsistency between space and property.

Response 3: Thanks very much for your valuable comments. As mentioned in your comments, the reason why industrial land use does not participate in the analysis of spatial heterogeneity is really unreasonable, and therefore we added this part. We analyzed the spatial heterogeneity of the influence of street centrality on the intensity of industrial land use, using geographically weighted regression model. (Table 3, Figure 9, Lines 528-541)

For details, please refer to the Word document.

Point 4: In addition, the paper does not explain why it is necessary or meaningful to choose four types of land use, and the dynamic mechanism between traffic network and land use intensity can be supplemented.

Response 4: Thank you very much for the suggestion. We chose these four types of land mainly based on the following consideration. In terms of the proportion of various types of land, residential land, industrial land, commercial land and public service land accounted for 30.47%, 17.55%, 13.02% and 12.27% respectively. The total amount of four types of land is 73.31%. Therefore, it can be considered that they represent the main land use types in the central urban area of Jinan.(Lines 179-185)

The dynamic mechanism analysis of street centrality and land use intensity in this paper really needs to be strengthened, which is only involved in the discussion part. Limited to the length of the paper and the availability of data, we did not discuss in depth. This is also what we need to focus on in the future. In the future, if we can obtain the data of traffic network and land use change in a long time series, we believe that we can draw meaningful conclusions in the dynamic mechanism analysis. Thank you again for your valuable suggestions on our research.

Point 5: Finally, in line 390, “four scenarios” should be replaced by “three scenarios”, because the line 388 mention “except the industrial land use”.

Response 5: Thank you very much for pointing out this error. “four scenarios” has been replaced by “three scenarios”. (Line 422)
